# The benefits and harms of open notes in mental health: A Delphi survey of international experts

**Charlotte Blease**[1]*, **Anna Kharko**[2], **Maria Hägglund**[3], **Stephen O'Neill**[1,4], **Deborah Wachenheim**[1], **Liz Salmi**[1], **Kendall Harcourt**[1], **Cosima Locher**[5,6], **Catherine M. DesRoches**[1,4], **John Torous**[4,7]

**1** General Medicine and Primary Care, Beth Israel Deaconess Medical Center, Boston, Massachusetts, United States of America, **2** Faculty of Health, University of Plymouth, Plymouth, United Kingdom, **3** Department of Women's and Children's Health, Uppsala University, Uppsala, Sweden, **4** Harvard Medical School, Boston, Massachusetts, United States of America, **5** Department of Clinical Psychology and Psychotherapy, University of Basel, Basel, Switzerland, **6** Department of Consultation-Liaison Psychiatry and Psychosomatic Medicine, University Hospital Zurich, Zurich, Switzerland, **7** Department of Psychiatry, Beth Israel Deaconess Medical Center, Boston, Massachusetts, United States of America

* cblease@bidmc.harvard.edu

**Data Availability Statement:** Data cannot be shared publicly because the Delphi poll is confidential to other respondents but the participants' responses were not anonymous to the

## Abstract

### Importance

As of April 5, 2021, as part of the 21st Century Cures Act, new federal rules in the U.S. mandate that providers offer patients access to their online clinical records.

### Objective

To solicit the view of an international panel of experts on the effects on mental health patients, including possible benefits and harms, of accessing their clinical notes.

### Design

An online 3-round Delphi poll.

### Setting

Online.

### Participants

International experts identified as clinicians, chief medical information officers, patient advocates, and informaticians with extensive experience and/or research knowledge about patient access to mental health notes.

### Main outcomes, and measures

An expert-generated consensus on the benefits and risks of sharing mental health notes with patients.

survey team. This is because Delphi polls use iterative techniques that involve recirculating bespoke surveys which include the earlier responses of each individual participant together with the aggregate response of the group.

**Funding:** Keane Scholar Award: CB Beyond Implementation of eHealth (2020-01229) funded by FORTE – the Swedish Research Council for Health, Working Life and Welfare: MH, CB, CD, KH. NORDeHEALTH (100477) funded by NordForsk: MH Gordon & Betty Moore Foundation: CD Cambia Health: CD

**Competing interests:** The authors have declared that no competing interests exist.

## Results

A total of 70 of 92 (76%) experts from 6 countries responded to Round 1. A qualitative review of responses yielded 88 distinct items: 42 potential benefits, and 48 potential harms. A total of 56 of 70 (80%) experts responded to Round 2, and 52 of 56 (93%) responded to Round 3. Consensus was reached on 65 of 88 (74%) of survey items. There was consensus that offering online access to mental health notes could enhance patients' understanding about their diagnosis, care plan, and rationale for treatments, and that access could enhance patient recall and sense of empowerment. Experts also agreed that blocking mental health notes could lead to greater harms including increased feelings of stigmatization. However, panelists predicted there could be an increase in patients demanding changes to their clinical notes, and that mental health clinicians would be less detailed/accurate in documentation.

## Conclusions and relevance

This iterative process of survey responses and ratings yielded consensus that there would be multiple benefits and few harms to patients from accessing their mental health notes. Questions remain about the impact of open notes on professional autonomy, and further empirical work into this practice innovation is warranted.

## Introduction

As of April 5, 2021, new federal rules in the U.S. mandate that all health providers offer patients online access to their clinical notes [1,2]. These new information sharing rules resulting from the 21st Century Cures Act of 2016 require rapid and full online access to test results, medication lists, referral information, and progress notes (so-called 'open notes'). Psychotherapy notes are exempt, and "information blocking" is permitted, if doing so ". . .will substantially reduce the risk of harm" where this is understood as physical harm to a patient or to another person or if there is a privacy exception (§ 171.201(a) p. 704) [1]. Licensed health professionals can decide what constitutes a substantial risk when working ". . .in the context of a current or prior clinician-patient relationship" (p. 702).

Sharing access to mental health notes is controversial, and in countries that have already opened notes to patients, many psychiatric organizations have resisted implementing the practice [3]. In surveys, mental health clinicians worry that patients will become anxious, confused or upset by what they read; many also anticipate access will lead to increased work burdens [4–6]. Findings drawn predominantly from primary care suggest that mental health patients may derive benefits from accessing their clinical notes including feeling more in control of their care, better remembering their care plan, and better understanding the rationale for medications [7,8]. However, sharing mental health notes could present more challenges than in other clinical specialties. Currently, only a few pilot surveys have examined patients' experiences in specialized mental health settings, and while the results are encouraging, at least some patients reported feeling disrespected or judged by what they read [9–12]. In existing studies, small sample sizes, the exclusion of patients with serious mental illnesses (SMI) or personality disorders, and the possibility of clinician and patient response biases also limit the potential for informative inferences about psychiatric care [13].

In light of limitations, clinicians may be uncertain about when it is appropriate to redact information. For example, it is unclear whether access may exacerbate harms among mental health patients. On the other hand, blocking information may deny patients important opportunities to better understand, manage, and become engaged in their care. Considering the pressing need for greater clarity in light of the new U.S. healthcare regulations taking effect in April 2021, and while awaiting further empirical research to inform best practice, our goal was to establish expert-led consensus about the potential benefits and harms of sharing notes in mental healthcare including among persons with serious and complex needs. Therefore, we set out to conduct the first Delphi poll in the field, encompassing a range of stakeholders including patient advocates [14].

## Methods

### Background

First developed by the Rand Corporation in the 1950s, Delphi polls are now an established qualitative methodology for exploring the consensus views of experts in an emerging field [15,16]. Delphi polls rely on non-probability sampling and studies show they offer more accurate predictions than other forecasting techniques [17–19]. The approach is designed to anonymously pool the views and predictions of a purposive sample of identified experts who are invited to answer a series of open-ended questions on a focused topic. Participants are next required to reassess their initial judgements, in light of aggregate group feedback, until consensus is obtained. While some Delphi polls employ the use of face-to-face meetings to help establish areas of agreement, more robust approaches employ anonymous, iterative techniques to avoid the influences of group think or dominant personalities [15,18]. Delphi polls therefore have clear advantages over focus groups owing to participants' anonymity and have been used extensively in healthcare policy and management, including in psychiatry [18,20,21].

### Approach

We used a modified Delphi technique structured around three discrete online surveys [22,23]. While there is no universal agreement about the sample size for Delphi polls, a key objective of Delphi is to maximize the response rate between each round which has been demonstrated to improve the accuracy of consensus opinions and forecasts [15,18,22]. In line with this approach, in Round 1, questions were open-ended requiring free-text responses which were then aggregated and translated into survey items (see S1 Appendix: Round One Survey for questions). In Round 2, experts were provided with the list of survey items and asked to rate their level of agreement using Likert scales. To reduce survey fatigue, those statements that reached a predefined level of consensus were omitted for Round 3, and remaining items recirculated. In this final round, participants were reminded of their previous judgment of each of the remaining items and furnished with the median response of the other experts and invited to preserve or revise their previous response.

### The expert panel

Using purposive sampling, the research team compiled a list of prospective participants with expertise on sharing mental health notes. We defined expertise as individuals with experience as: clinicians sharing mental health notes with patients; patients with mental health diagnoses, including patient advocates with lived experience or knowledge of the practice; chief medical information officers or directors of divisions of health organizations that offered open notes to mental health patients; and/or, academic informaticians, who had published significant

contributions within the field of health informatics and patient access to clinical records including authors of publications skeptical about the innovation. The research team was mindful to ensure gender, age, race, and international diversity, and specifically strove to include representation from countries and health systems where open notes have been implemented. To ensure a diversity of representation we also employed a snowballing technique by inviting chief medical information officers of health organizations that had opened psychiatric notes to assist us by providing names and contact details of mental health clinicians with experience of the practice. The final list of experts included 92 prospective participants. The study received approval from the Beth Israel Deaconess Medical Center Institutional Review Board in April 2020 (Protocol # 2020P000218) and the University of Plymouth, UK (Protocol # 19/20-1331). Invited participants were advised that the survey was voluntary and unpaid, that their responses would be anonymous to other participants, and that their identity was restricted to members of the research team. Identified experts were contacted via email in August 2020 with an invitation and link to the web-based survey. Invitees were informed that an adequate response time would be given between rounds and that they could choose to withdraw at any time. All respondents gave informed consent before participating.

## The questionnaire

An electronic survey was created using Qualtrics (Qualtrics, Provo, UT). Since all prospective participants were fluent English speakers, the survey was only administered in this language. The poll incorporated the three-step modified Delphi technique, and data collection ran from August to November 2020. Participants were sent up to 4 reminders, 1 week apart, and given 4 to 5 weeks to respond to each round (see Fig 1, Flowchart of Delphi Poll). The first round comprised demographic questions and requested information about the nature of participants' expertise with open notes. This was followed by six open-ended questions on sharing mental health notes plus one additional open-ended question allowing participants to comment on the survey or submit additional responses (see S1 Appendix: Round One Survey). The present Delphi study was based on respondents' answers to two questions on the potential benefits and harms to patients of reading mental health notes. Participants were asked: "What, in your opinion, are the benefits–if any–of sharing mental health notes with patients?" and "What, in your opinion are the harms–if any–of sharing mental health notes with patients?" Answers to other questions formed the basis of a separate qualitative study that has been published elsewhere.

Following the closure of Round 1, descriptive content analysis was used to transform responses into lists of statements [24]. Owing to the limitations of the data-set (brief comments or sentence fragments), full thematic coding was not applicable [25]. Coding was conducted by CB and independently reviewed by JT and MH and subsequent refinements were made. These individuals were selected for their diverse epistemological backgrounds: CB is a philosopher of medicine and healthcare ethicist from the UK, JT is a psychiatrist with experience of sharing mental health notes with patients in the US, and MH is a healthcare informaticist and implementation scientist based in Sweden. Comments that were unclear or deemed irrelevant to the question were excluded. Statements were then transformed into survey items, and where possible participants' exact phrasing was preserved. Survey items relating to benefits, and harms of sharing mental health notes formed two separate sections of the survey circulated in Round 2, and participants were requested to respond to each question using a variety of predefined 7-point Likert scales: 1 = *strongly disagree*, 2 = *moderately disagree*, 3 = *slightly disagree*, 4 = *neutral*, 5 = *slightly agree*, 6 = *moderately agree*, 7 = *strongly agree* (see S2 Appendix: Round Two Survey). Prior to analyzing responses to Rounds 2 and 3, consensus to items was set at an

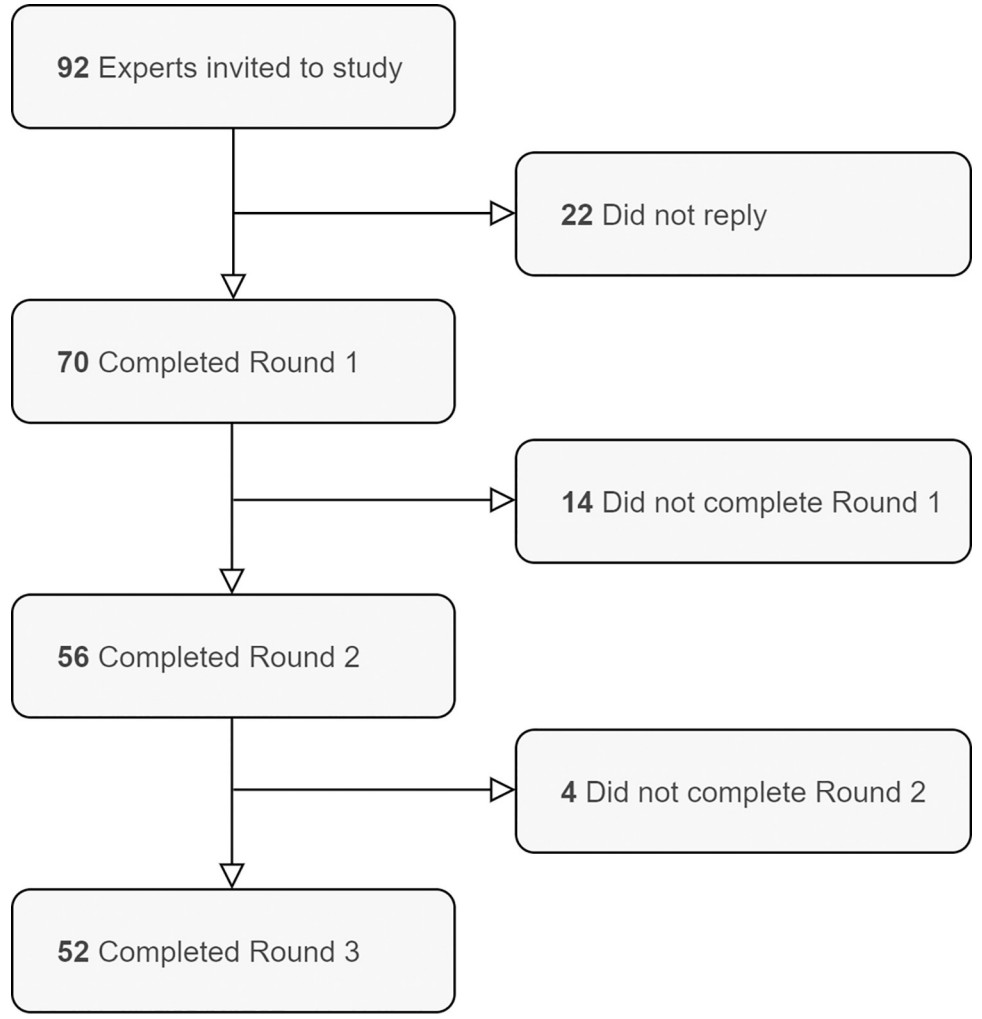

**Fig 1. Flowchart of Delphi poll.**

interquartile range of ≤ 1 [15,26]. After analysis of Round 2 responses, items that did not reach consensus were recirculated in Round 3. In Round 3, each participant received a bespoke survey in which they were reminded of their responses to items in Round 2 and provided with the median response of all participants. After competing closed-ended survey questions in each round, participants were also invited to provide any additional feedback. All survey items that reached consensus were collated into themes, and subthemes by CB and JT.

## Results

### Overview

A total of 70 of 92 (76%) participants from 6 countries responded to Round 1. Among respondents, 35 of 70 (50%) identified as male, and the mean age was 50 (see Table 1). Forty-seven of 70 (67%) reported currently working in clinical practice, and 7 of 70 (10%) reported lived experience of illness. The mean number of years of experience working as a clinician, an open notes researcher, or as a patient advocate was 16 years. All respondents left comments (5,760 words) which were typically brief (1 or 2 sentences or sentence fragments. Replies were shorter

**Table 1. Characteristics of survey respondents by round.**

| | No. (%) of Respondents | | |
|---|---|---|---|
| Characteristic | Round 1 (n = 70) | Round 2 (n = 56 of 70 [80.0%]) | Round 3 (n = 52 of 56 [92.8%]) |
| Gender | | | |
| Male [a] | 35 (50) | 28 (50) | 26 (50) |
| Age range, years | | | |
| 20–29 | 1 (1) | - | - |
| 30–39 | 19 (27) | 17 (30) | 17 (33) |
| 40–49 | 14 (20) | 11 (20) | 10 (19) |
| 50–59 | 19 (27) | 15 (27) | 14 (27) |
| $\geq 60$ | 17 (24) | 13 (23) | 11 (21) |
| Ethnicity | | | |
| Asian | 6 (9) | 4 (7) | 4 (8) |
| Black/African/Caribbean | 1 (1) | 1 (2) | 1 (2) |
| White | 59 (84) | 49 (88) | 46 (88) |
| Other | 2 (3) | 1 (2) | - |
| Declined to answer | 2 (3) | 1 (2) | 1 (2) |
| Country of residence | | | |
| Canada | 3 (4) | 2 (4) | 2 (4) |
| Estonia | 1 (1) | 1 (2) | 1 (2) |
| Norway | 3 (4) | 3 (5) | 3 (6) |
| Sweden | 12 (17) | 10 (18) | 10 (19) |
| UK | 4 (6) | 4 (7) | 4 (8) |
| USA | 47 (67) | 36 (64) | 32 (62) |
| Currently works in clinical practice | 45 (64) | 34 (61) | 31 (60) |
| Psychiatry | 15 (21) | 10 (18) | 9 (17) |
| Primary care | 10 (14) | 8 (14) | 8 (15) |
| Clinical psychology/Psychotherapy | 7 (10) | 6 (11) | 6 (12) |
| Psychiatric Nursing | 4 (6) | 4 (7) | 4 (8) |
| Paediatrics | 3 (4) | 3 (5) | 2 (4) |
| Social work | 3 (4) | 1 (2) | - |
| Palliative care/Home hospice | 2 (3) | 1 (2) | 1 (2) |
| Peer support | 2 (3) | 2 (4) | 2 (4) |
| Hospitalist | 1 (1) | 1 (2) | 1 (2) |
| Radiology | 1 (1) | 1 (2) | 1 (2) |
| Occupation or expertise related to health/open notes informatics research/patient advocacy [c] | | | |
| Clinician | 46 (66) | 35 (63) | 32 (62) |
| Researcher | 25 (31) | 21 (38) | 21 (40) |
| Chief Information Officer/Portal Director/Medical Director | 14 (20) | 12 (21) | 9 (17) |
| Patient Advocate/Person with Lived Experience | 5 (7) | 4 (7) | 4 (8) |
| Social Worker | 1 (1) | 1 (2) | 1 (2) |

[a] Including one transgender male.

[b] Including Adult, Child and Adolescent Psychiatry.

[c] Numbers & percentages may add up greater than the total n, as some participants reported multiple expertise.

when discussing benefits (2,639 words) than harms (3,121 words). As a result of descriptive analysis of free-text responses, the survey was expanded into 88 items, 42 items on the potential benefits of sharing mental health notes with patients and 46 on potential harms (see S2 Appendix: Round Two Survey).

In Round 2, 56 of 70 (80%) experts participated in the online survey and reached consensus on 22 of 88 items (25%) (see S3 Appendix: De-identified Raw Data Results of Round Two). The mean age was 50 years old, and the mean number of years of experience working as a clinician, open notes researcher, or as a patient advocate was 17 years. In Round 3, 52 of 56 (93%) experts responded, reaching consensus on a further 43 items. In total, consensus was achieved for 65 of 88 items (74%). Among Round 3 respondents, the mean age was 49, and the mean number of years of experience working as a clinician, an open notes researcher, or as a patient advocate was 17 years. Table 2 presents the items that reached consensus after each round. Following thematic analysis, all items that reached consensus were categorized into four themes; items in shaded boxes in Table 2 represent perceived benefits of open notes in mental healthcare.

**Table 2. Items on the benefits and harms of open notes in mental healthcare that reached consensus.**

| Item | Mean (SD) Rating | Median–Consensus [a] |
|---|---|---|
| **Effects on Patient Understanding, Recall and Empowerment (20 items)** | | |
| Access to mental health notes will improve patient recall about what was communicated during the visit. | 6.3 (.89) | 7 –Strongly agree |
| Access to mental health notes will improve patient recall especially among patients with memory or cognitive problems. | 6.2 (.77) | 6 –Moderately agree |
| Access to mental health notes will improve patient recall of homework between visits. | 6.02 (.96) | 6 –Moderately agree |
| Access to mental health notes will improve patient understanding about their care plan. | 6 (.86) | 6 –Moderately agree |
| Access to mental health notes will improve patient understanding about their diagnosis/mental health condition. | 5.85 (.89) | 6 –Moderately agree |
| Access to mental health notes will improve patient understanding about the rationale for their treatments. | 5.75 (.95) | 6 –Moderately agree |
| Access to mental health notes will improve patient understanding about their health changes over time. | 5.71 (1.07) | 6 –Moderately agree |
| Access to mental health notes will improve patient awareness over current medications. | 5.77 (.96) | 6 –Moderately agree |
| Access to mental health notes will improve patient preparation for visits. | 5.46 (1.16) | 6 –Moderately agree |
| Access to mental health notes will increase the likelihood patients will ask relevant questions at visits. | 5.38 (1.14) | 6 –Moderately agree |
| Patient access to mental health notes will improve patient-clinician communication. | 5.75 (1.03) | 6 –Moderately agree |
| Patient access to mental health notes will help to demystify psychotherapy. | 5.37 (1.17) | 6 –Moderately agree |
| Access to mental health notes will improve patient sense of control of their health. | 5.75 (.99) | 6 –Moderately agree |
| Patient access to mental health notes will prompt patients to research more about their healthcare. | 5.27 (1.01) | 5 –Somewhat agree |
| Access to mental health notes will improve patient sense of privacy over their health information. | 4.6 (1.19) | 5 –Somewhat agree |
| Access to mental health notes will improve patient responsibility for their healthcare. | 5.18 (.99) | 5 –Somewhat agree |

*(Continued)*

**Table 2.** (Continued)

| Item | Mean (SD) Rating | Median–Consensus [a] |
|---|---|---|
| Patient access to mental health notes will improve family/friend/caregiver recall about patient appointments. | 5.45 (1.06) | 5 –Somewhat agree |
| When accessing their mental health notes patients will be confused by clinical abbreviations. | 5.25 (1.56) | 5 –Somewhat agree |
| When accessing their mental health notes patients will be confused by psychiatric terms. | 4.65 (1.31) | 5 –Somewhat agree |
| Patients in domestic abuse situations will be less candid in visits. | 4.65 (1.1) | 5 –Somewhat agree |
| **Relational Effects (17 items)** | | |
| Patient access to mental health notes will improve mutual understanding between patients and clinicians. | 5.79 (.98) | 6 –Moderately agree |
| Patient access to mental health notes will improve patient-clinician goal-alignment. | 5.75 (1.05) | 6 –Moderately agree |
| Access to mental health notes will increase adoption of patient-centered language in clinicians' documentation. | 5.67 (1.13) | 6 –Moderately agree |
| Patient access to mental health notes will improve the therapeutic alliance. | 5.45 (1.11) | 6 –Moderately agree |
| Access to mental health notes will diminish the use of disrespectful or derogatory language in notes. | 5.96 (.95) | 6 –Moderately agree |
| Access to mental health notes will diminish patient anxiety about what clinicians write about them. | 5.6 (1.11) | 6 –Moderately agree |
| Access to mental health notes will increase shared decision-making. | 5.52 (.95) | 6 –Moderately agree |
| Access to mental health notes will increase the depth of patient-clinician dialogue during visits. | 5.13 (1.07) | 5 –Somewhat agree |
| Access to mental health notes will improve patient trust in clinicians. | 5.55 (1.01) | 6 –Moderately agree |
| Access to mental health notes will improve patient satisfaction with care. | 5.23 (1.08) | 5 –Somewhat agree |
| Patient access to mental health notes will improve family/friend/caregiver trust in clinicians. | 5 (1.01) | 5 –Somewhat agree |
| As a result of reading their mental health notes, a minority of patients will feel ashamed. | 4.71 (1.07) | 5 –Somewhat agree |
| As a result of reading their mental health notes, a minority of patients will feel insulted. | 4.80 (1.29) | 5 –Somewhat agree |
| As a result of reading their mental health notes, a minority of patients will feel alienated by clinical language. | 4.86 (1.26) | 5 –Somewhat agree |
| Perceived misunderstandings in mental health notes will cause patients to be frustrated. | 5.11 (.93) | 5 –Somewhat agree |
| Perceived misunderstandings in mental health notes will cause patients to be angry. | 4.84 (.95) | 5 –Somewhat agree |
| Access to mental health notes will be detrimental to the therapeutic alliance. | 2.5 (1.36) | 2 –Moderately disagree |
| **Quality of Care, Outcomes, and Patient Safety (22 items)** | | |
| Hiding mental health/psychotherapy notes will lead to greater patient stigmatization. | 5.56 (.98) | 6 –Moderately agree |
| Patient access to mental health notes will help to close the feedback loop on care. | 5.56 (.98) | 6 –Moderately agree |
| Access to mental health notes will help patients to correct errors in notes. | 5.37 (1.58) | 6 –Moderately agree |
| Access to mental health notes will help patients to correct clinician misinterpretations | 5.35 (1.62) | 6 –Moderately agree |

(*Continued*)

**Table 2.** (Continued)

| Item | Mean (SD) Rating | Median–Consensus [a] |
|---|---|---|
| Patient access to mental health notes will improve quality of care. | 5.34 (1.15) | 5 –Somewhat agree |
| Patient access to mental health notes will improve treatment processes. | 5.34 (1.16) | 5 –Somewhat agree |
| Patient access to mental health notes will improve clinical outcomes. | 5.02 (1.06) | 5 –Somewhat agree |
| Access to mental health notes will improve patient adherence to treatment plans. | 5.24 (1.07) | 5 –Somewhat agree |
| Access to mental health notes will improve patient adherence to medications. | 5.12 (.96) | 5 –Somewhat agree |
| As a result of opening mental health notes there will be harms if notes are written without intention to share with patients. | 5.39 (1.04) | 5 –Somewhat agree |
| As a result of opening mental health notes there will be harms if patients lack guidance on reading notes. | 4.52 (1.32) | 5 –Somewhat agree |
| As a result of reading their mental health notes, a minority of patients will be distressed. | 5.16 (1.02) | 5 –Somewhat agree |
| Access to mental health notes will diminish patient feelings of stigmatization. | 4.75 (1.19) | 5 –Somewhat agree |
| As a result of reading their mental health notes, a minority of patients will experience cyberchondria. | 4.43 (1.26) | 4 –Neutral |
| Patients in domestic abuse situations will be at increased harm. | 4.27 (1.05) | 4 –Neutral |
| Too much detail in notes will re-traumatize patients. | 3.48 (1.21) | 4 –Neutral |
| Access to mental health notes will be detrimental to patient adherence to treatment plans. | 2.29 (1.14) | 2 –Moderately disagree |
| Patients with the following conditions/symptoms will be harmed from accessing their notes: bipolar disorder disorders. | 2.31 (1.08) | 2 –Moderately disagree |
| Patients with the following conditions/symptoms will be harmed from accessing their notes: personality disorders. | 2.6 (1.32) | 2 –Moderately disagree |
| Patients with the following conditions/symptoms will be harmed from accessing their notes: patients who are suicidal. | 2.35 (1.08) | 2 –Moderately disagree |
| Patients with the following conditions/symptoms will be harmed from accessing their notes: patients with obsessive conditions. | 2.35 (1.12) | 2 –Moderately disagree |
| Patients with the following conditions/symptoms will be harmed from accessing their notes: eating disorders. | 2.33 (1.2) | 2 –Moderately disagree |
| **Effects on Professional Autonomy & Healthcare Efficiencies (6 items)** | | |
| Clinicians will be less detailed/accurate in documenting negative aspects of patient relationship. | 4.93 (1.14) | 5 –Somewhat agree |
| Clinicians will be less detailed/accurate in documenting patients' personalities. | 4.77 (1.25) | 5 –Somewhat agree |
| As a result of opening mental health notes there will be an increase in patients demanding changes to their notes. | 4.73 (1.29) | 5 –Somewhat agree |
| Access to mental health notes will increase efficiency in patient care. | 4.56 (1.3) | 5 –Somewhat agree |
| Clinicians will spend less time in patient visits. | 2.69 (.98) | 3 –Somewhat disagree |
| Access to mental health notes will be detrimental to clinician honesty in therapy sessions. | 2.69 (1.08) | 3 –Somewhat disagree |

Abbreviations: IQR–interquartile range.

[a] Response scale included the following levels: 1 –Strongly disagree, 2 –Moderately disagree, 3 –Somewhat disagree, 4 –Neutral, 5 –Somewhat agree, 6 –Moderately agree, 7 –Strongly agree.

Shaded items represent perceived benefits of opening mental health notes.

### Themes

**Effects on patient understanding, recall, and empowerment.**    Panelists agreed that open notes could aid patient recall about what was communicated in visits, and that access to documentation could improve patient-clinician communication, including patient understanding about their diagnosis, prescribed medications, and the rationale for treatments. They also agreed that access could improve mental health patients' sense of control over their healthcare, insight about their health, and help them better prepare for visits.

**Relational effects.**    Reflecting on whether access could affect the patient-clinician relationship, there was agreement among experts that open notes could improve trust, enhance the therapeutic alliance, increase patient-clinician goal-alignment, and strengthen shared decision-making. Panelists also believed that patient access would increase the adoption of patient-centered language and diminish the use of disrespectful or derogatory language in documentation. There was also agreement that access to clinical notes could diminish patients' anxiety about what clinicians might write about them.

**Quality of care, outcomes, and patient safety.**    Commenting on the potential for adverse effects, panelists *disagreed* that open notes would be harmful to patients with serious mental health diagnoses, specifically including persons with bipolar disorders, major depressive disorders, personality disorders, eating disorders, obsessive conditions, and individuals who are suicidal. Panelists agreed that access could help patients to correct errors in notes, including clinician misinterpretations, and that offering access could help to close the feedback loop on care. Furthermore, there was consensus that hiding mental health or psychotherapy notes might lead to greater patient stigmatization or harm. However, participants "somewhat agreed" that harms could arise if clinicians wrote notes without the intention that they might be read by patients, or if patients lacked guidance on how to read their notes. Panelists also "somewhat agreed" that access could improve treatment, including medication adherence.

**Effects on professional autonomy and healthcare efficiencies.**    Expert panelists "somewhat agreed" that as a result of access, there could be an increase in patients demanding changes to their clinical notes. Commenting on the effects on professional autonomy, panelists also "somewhat agreed" that mental health clinicians would be less detailed/accurate in documenting negative aspects of the patient relationship, details about patients' personalities, or symptoms of paranoia in patients. However, experts predicted that patient access to mental health notes could increase efficiency in care delivery.

## Discussion

Currently, limited attention has been paid to the benefits and harms of reading mental health notes among patients with psychiatric diagnoses, including persons with SMI. In view of the lack of large-scale studies into the effects on mental health patients' experiences, we used a rigorous modified Delphi approach to establish the consensus views about this practice innovation among an international stakeholder group of clinicians, patients, chief medical information officers, and informaticians. The final results comprise the first consensus-driven statement on the benefits and harms of online access to mental health notes.

In this Delphi poll there was consensus that offering online access to mental health notes could enhance patients' understanding about their diagnosis, care plan, and rationale for treatments. These views contrast with findings of small scale surveys in the U.S. and Sweden in which mental health clinicians anticipated most patients would find notes more confusing than helpful [4,5]. There was also consensus that access could enhance patient recall and sense of empowerment about their care plan. The expert panel also agreed that patient access to

mental health notes could strengthen multiple relational benefits of care. The panel believed that open notes present no special harms to patients with SMI.

Despite clear consensus on the potential benefits to patient autonomy questions were raised about the effects on professional autonomy. Experts anticipated changes to the detail and accuracy of records with respect to information patients might perceive as negative. Notably, while experts expect patients to request changes to their notes, they also believed that patient feedback could improve documentation quality. Finally, there was consensus that blocking mental health notes could lead to greater harms including increased feelings of stigmatization. Small scale qualitative studies in psychotherapy suggest that at least some patients experience negative feelings as a consequence of being denied access to their notes [11,12].

The present study invites a number of important unresolved questions. It was not established whether the panelists believed that any potential changes in the detail or accuracy of records might diminish the quality of care. Perhaps reflecting lack of research on key issues, expert panelists held no strong views about whether patients who disagreed with their diagnosis would be less likely to attend visits or whether too much detail might retraumatize patients. Our expert panelists also reported "neutral" consensus opinions on whether clinicians would be less detailed/accurate in writing differential diagnoses or in documenting substance abuse disorders. These and other issues, warrant further, more nuanced empirical examination and we strongly recommend that research investigate patients' experiences with accessing mental health notes across a range of settings, including outpatient and inpatient care, and encompassing a wide range of patient populations with different mental health diagnoses. Focused empirical research, including randomized controlled trials, are required to better understand whether access influences objective health outcomes and attendance at visits, and whether patients feel judged or offended by what they read [27]. There is a need for data among patients with serious mental illness, including schizophrenia, personality disorders, or active suicidality to avoid overgeneralizing current evidence to these populations and use cases. Larger scale studies are needed to examine psychiatric clinicians' experiences including emergent concerns with the practice, and potential sources of patient-clinician disagreement. It will be important to determine objective changes to documentation as a result of patient access [28]. Combined, empirical research should aim to inform how the innovation might be optimized for its new dual purpose–both as a detailed aide memoire for clinicians and as a tool for communication with patients. Finally, it remained unresolved when redaction of aspects of mental records from patients might be appropriate [29], and future research should elaborate on how best this can be practically and ethically managed for both patients and clinicians [30].

## Limitations

This study has several limitations many of which are inherent to the Delphi methodology. As with all Delphi polls, there are no standard guidelines for identifying expertise. Although we selected participants to represent a diverse spectrum comprising healthcare professionals, patients with lived experiences of mental illnesses, and health informatics researchers, the reliability of consensus opinions in this survey is dependent on the specialist knowledge and experiences of those who participated. While our expert panel represented a diversity of expertise from countries and health organizations where patients are currently offered access to their notes, it is likely that the survey would have benefited from greater ethnic and socio-economic diversity. Notably, challenges related to portal access, and the digital divide, received less attention, and this may have been a consequence of the demographic composition of panelists. In addition, we cannot exclude the possibility that, even though clinicians and chief medical

information officers had extensive knowledge and experience of opening mental health notes, some individuals might already have been more positively disposed to the practice, influencing results. Notwithstanding these limitations, the study benefited from high response rates between each round.

## Conclusions

In the era of health information transparency, both patient and professional autonomy in mental health contexts must be balanced with the potential benefits and risks to patient care. Experts in this Delphi poll anticipated multiple benefits and few harms of patient access to mental health notes. Further empirical inquiry is required to explore the impact on both patients and clinicians of psychiatric patients of reading their notes, and how both patients and clinicians can become better prepared and supported for documentation transparency in mental healthcare.

## Supporting information

**S1 Appendix. Round one survey.**
(PDF)

**S2 Appendix. Round two survey.**
(PDF)

**S3 Appendix. De-identified raw data results of round two.**
(CSV)

## Acknowledgments

The authors express their gratitude to the experts who participated in this survey.

## Author Contributions

**Conceptualization:** Charlotte Blease, Anna Kharko, Catherine M. DesRoches, John Torous.

**Data curation:** Anna Kharko.

**Formal analysis:** Charlotte Blease, Anna Kharko, Maria Hägglund, Cosima Locher, John Torous.

**Funding acquisition:** Maria Hägglund, Catherine M. DesRoches.

**Investigation:** Charlotte Blease, Anna Kharko, Maria Hägglund, Stephen O'Neill, Deborah Wachenheim, Liz Salmi, Kendall Harcourt, John Torous.

**Methodology:** Charlotte Blease, Anna Kharko, Maria Hägglund, Stephen O'Neill, Deborah Wachenheim, Liz Salmi, Kendall Harcourt, John Torous.

**Project administration:** Charlotte Blease, Anna Kharko.

**Software:** Anna Kharko.

**Supervision:** Catherine M. DesRoches.

**Visualization:** Charlotte Blease.

**Writing – original draft:** Charlotte Blease.

**Writing – review & editing:** Charlotte Blease, Anna Kharko, Maria Hägglund, Stephen
O'Neill, Deborah Wachenheim, Liz Salmi, Kendall Harcourt, Cosima Locher, Catherine M.
DesRoches, John Torous.

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
