## [Decision Letter · Decision Letter 0]

6 Aug 2021

PONE-D-21-12428

The Benefits and Harms of Open Notes in Mental Health: A Delphi Survey of International Experts

PLOS ONE

Dear Dr. Blease,

Thank you for submitting your manuscript to PLOS ONE. After careful consideration, we feel that it has merit but does not fully meet PLOS ONE’s publication criteria as it currently stands. Therefore, we invite you to submit a revised version of the manuscript that addresses the points raised during the review process.

We look forward to receiving your revised manuscript.

Kind regards,

Umberto Volpe

Academic Editor

PLOS ONE

Journal Requirements:

2. Please include a caption for figure 1.

Additional Editor Comments (if provided):

As highlighted by referees, minor corrections should be applied in a revised version.

Reviewers' comments:

Reviewer's Responses to Questions

**Comments to the Author**

1. Is the manuscript technically sound, and do the data support the conclusions?

Reviewer #1: Yes

Reviewer #2: Yes

Reviewer #3: Yes

2. Has the statistical analysis been performed appropriately and rigorously? 

Reviewer #1: Yes

Reviewer #2: Yes

Reviewer #3: N/A

3. Have the authors made all data underlying the findings in their manuscript fully available?

Reviewer #1: Yes

Reviewer #2: Yes

Reviewer #3: No

4. Is the manuscript presented in an intelligible fashion and written in standard English?

Reviewer #1: Yes

Reviewer #2: Yes

Reviewer #3: Yes

5. Review Comments to the Author

Reviewer #1: This is a great paper and goes a long way to answer questions that have been raised by clinicians and IT leads for many years. Interestingly in a UK family practice that has 80% of patients able to access their records the highest percentage of patients accessing their records when we looked at this 5 years or so ago were patients with mental health problems. This suggests that patients with mental health issues d want t see their notes. The results of the questionnaires from an international group of specialists within countries that already practice patient access to records seem very reassuring and valuable. A small non peer reviewed study of ours https://bjgp.org/content/68/suppl_1/bjgp18X697133 showed that patients increased their knowledge of their own medical history when tested before and after seeing their notes. The benefits and harms of pen Notes in mental health paper is very timely. The very positive feelings of the interviewed experts would suggest that it might be unethical not to undertake a controlled trial of patients with mental health accessing their mental health records and measuring the patients response to the access against the 80 or so tested fields in the questionnaire.

Reviewer #2: This is an interesting paper and seems to me worthy of publication after some revision. The main problem for me was the abstract – which seemed lacking in actual results. In the abstract the setting seems to me to be methods. Should there not be a methods section in Abstract?. Setting = online.

Results in the Abstract is purely on the numbers responding rather than the content of the statements- benefits and harms. The only results come in the conclusions. Is there not space to put results in results? For example, that 88 items (42 potential benefits, 46 potential harms) were identified is a step more towards ‘results’- but better still to know what these were – summarising Table 2.

Approach (p5)

This is written in present tense – assuming that this is what was actually done, it should be in past tense- otherwise you are left with a feeling of doubt as to whether or not this is what the methods actually were. An indication earlier in the text that the questions were shown in Appendix 1 would be useful – I was left wondering what they were until later in the paper.

Results

It would help the reader if some ways was found of making where there was agreement on benefit and where on harm in Table 2. Given that some statements are worded in the positive and some in the negative, and then there is agreement or disagreement, it is not immediately clear where there was agreement on benefit or harm. You come on to this on page 15, but I have already been trying to work it out from the table – so a clearer table would help.

(p17) I find this sentence a bit odd – “Whether the panel believed any such changes might diminish the quality of care, however, was not established.” Isn’t it the case that there was no consensus? Ie opinion was divided? So isn’t it clearer to say (something like) Opinions of the panel were divided on whether quality of care might be diminished’. And so is this not just another of the ‘unresolved questions’ ie it should go into the next paragraph.

Reviewer #3: I found this paper interesting to read but would have liked to have seen more detail on the methodology. I also have concerns that the expert panel was not representative in that it consists of experts more likely to hold positive views about open notes. I wonder if the findings would have been very different if a more representative expert panel had been used.

Minor points

P1, lines 7, 9 and 11 – I would put a comma after ‘round 1’, ‘round 2’, and ‘round 3’.

P3, para 2, line 3 – I would put a comma after ‘In surveys’.

P4, line 4 – I wonder if the term ‘redact’ might come across as less judgemental than ‘hide’?

P5, para 2, line 3 – replace ‘clinicians of sharing mental health notes’ with ‘clinicians sharing mental health

notes’

P6, para 2, line 2. Would languages other than English have been made available if required?

P9, Table 1. Could “Age range, y” be replaced with “Age range, years” ?

P15, 4th line from bottom – as before would ‘redacting’ come across as less judgemental than ‘hiding’?

P16, line 2 – replace “medication, adherence” with “medication adherence”

Specific concerns

P5, ‘The expert panel’ – The authors describe how individuals with expertise included clinicians with experience of sharing mental health notes with patients and CMIOs of organizations that offer open notes to patients. This would seem to have introduced a bias into the sample in that clinicians and CMIOs who may have actively decided not to offer open notes were not represented?

P11, last sentence. The authors mention that thematic analysis was employed, on p7 the authors reference Braun and Clarke. I would like to know a little more information about the epistemological perspective taken by the authors and to see some evidence of reflexivity. I feel there could be more detail on the thematic analysis – the authors mention that 2 individuals were responsible for identifying themes and subthemes – how was this done and what procedures were in place when these individuals disagreed?

Pages 12-14. Whilst I can see the value of a thematic analysis of the questions generated by the expert panel, the data would seem to lend itself quite well to a factor analysis. Did the authors consider a factor analysis and if so, what were their reasons for not conducting one?

P16 – It would be good to see some consideration of the difficulties inherent in having two versions of a record when elements of the record have been redacted for the patient view. How can this be managed in practical terms so that clinicians can still see the full record?

6. PLOS authors have the option to publish the peer review history of their article (what does this mean?). If published, this will include your full peer review and any attached files.

Reviewer #1: **Yes: **Richard Fitton

Reviewer #2: **Yes: **Ray Jones

Reviewer #3: No

---

## [Author Response · Author response to Decision Letter 0]

19 Aug 2021

PONE-D-21-12428

The Benefits and Harms of Open Notes in Mental Health: A Delphi Survey of International Experts

PLOS ONE

Dear PLOS ONE,

Thank you for inviting a revised submission of our paper. We detail our responses to the reviewers below, and our grateful for their valuable feedback.

Yours

C. Blease, PhD

Reviewers' comments:

Reviewer's Responses to Questions

Comments to the Author

1. Is the manuscript technically sound, and do the data support the conclusions?

Reviewer #1: Yes

Reviewer #2: Yes

Reviewer #3: Yes

2. Has the statistical analysis been performed appropriately and rigorously?

Reviewer #1: Yes

Reviewer #2: Yes

Reviewer #3: N/A

3. Have the authors made all data underlying the findings in their manuscript fully available?

Reviewer #1: Yes

Reviewer #2: Yes

Reviewer #3: No

4. Is the manuscript presented in an intelligible fashion and written in standard English?

Reviewer #1: Yes

Reviewer #2: Yes

Reviewer #3: Yes

5. Review Comments to the Author

Please use the space provided to explain your answers to the questions above. You may also include additional comments for the author, including concerns about dual publication, research ethics, or publication ethics. (Please upload your review as an attachment if it exceeds 20,000 characters).

Reviewer #1: 

This is a great paper and goes a long way to answer questions that have been raised by clinicians and IT leads for many years. Interestingly in a UK family practice that has 80% of patients able to access their records the highest percentage of patients accessing their records when we looked at this 5 years or so ago were patients with mental health problems. This suggests that patients with mental health issues d want t see their notes. The results of the questionnaires from an international group of specialists within countries that already practice patient access to records seem very reassuring and valuable. A small non peer reviewed study of ours https://bjgp.org/content/68/suppl_1/bjgp18X697133 showed that patients increased their knowledge of their own medical history when tested before and after seeing their notes. The benefits and harms of pen Notes in mental health paper is very timely. The very positive feelings of the interviewed experts would suggest that it might be unethical not to undertake a controlled trial of patients with mental health accessing their mental health records and measuring the patients response to the access against the 80 or so tested fields in the questionnaire.

Response: We thank the reviewer for these kind comments and feedback. The reviewer raises an excellent point about RCTs which we have now added in the Discussion on pages 18-19:

“Focused empirical research, including randomized controlled trials, are required to better understand whether access influences objective health outcomes and attendance at visits, and whether patients feel judged or offended by what they read.[27]”

Reviewer #2: 

This is an interesting paper and seems to me worthy of publication after some revision. The main problem for me was the abstract – which seemed lacking in actual results. In the abstract the setting seems to me to be methods. Should there not be a methods section in Abstract?. Setting = online. Results in the Abstract is purely on the numbers responding rather than the content of the statements- benefits and harms. The only results come in the conclusions. Is there not space to put results in results? For example, that 88 items (42 potential benefits, 46 potential harms) were identified is a step more towards ‘results’- but better still to know what these were – summarising Table 2.

Response: We thank the reviewer for this helpful feedback. We have now amended the Abstract (which is indeed restricted by the word count). We have now amended Setting to “Online” and added more detailed information on the actual results: 

“A total of 70 of 92 (76%) experts from 6 countries responded to Round 1. A qualitative review of responses yielded 88 distinct items: 42 potential benefits, and 48 potential harms. A total of 56 of 70 (80%) experts responded to Round 2, and 52 of 56 (93%) responded to Round 3. Consensus was reached on 65 of 88 (74%) of survey items. There was consensus that offering online access to mental health notes could enhance patients’ understanding about their diagnosis, care plan, and rationale for treatments, that access could enhance patient recall and sense of empowerment. Experts also agreed that blocking mental health notes could lead to greater harms including increased feelings of stigmatization. However, panelists predicted there could be an increase in patients demanding changes to their clinical notes, and that mental health clinicians would be less detailed/accurate in documentation.”

Approach (p5)

This is written in present tense – assuming that this is what was actually done, it should be in past tense- otherwise you are left with a feeling of doubt as to whether or not this is what the methods actually were. An indication earlier in the text that the questions were shown in Appendix 1 would be useful – I was left wondering what they were until later in the paper.

Response: The Approach and Methods section have now amended to past tense, and reference to Appendix 1 has also been signposted earlier in the text.

Results

It would help the reader if some ways was found of making where there was agreement on benefit and where on harm in Table 2. Given that some statements are worded in the positive and some in the negative, and then there is agreement or disagreement, it is not immediately clear where there was agreement on benefit or harm. You come on to this on page 15, but I have already been trying to work it out from the table – so a clearer table would help.

Response: We do agree that the table is, in some respects, not intuitive. However, we followed other Delphi studies in the layout which lists items according to strength of opinion reached in the consensus. In addition, as with other Delphi polls, we have preserved the survey items, as far as possible, according to the wording used by the participants in Round 1. Nonetheless, the the reviewer raises an important point and in order for items about harms and benefits to be clearer in Table 2, and more intuitive for the reader, we have now shaded those items that represent perceived benefits of opening mental health notes. Please see pages 13-14. 

(p17) I find this sentence a bit odd – “Whether the panel believed any such changes might diminish the quality of care, however, was not established.” Isn’t it the case that there was no consensus? Ie opinion was divided? So isn’t it clearer to say (something like) Opinions of the panel were divided on whether quality of care might be diminished’. And so is this not just another of the ‘unresolved questions’ ie it should go into the next paragraph.

Response: We have now moved this sentence to the next paragraph. In fact, the original phrasing was accurate and unavoidably nuanced since it was not established (one way or the other) from the Delphi poll whether participants did believe that any changes to documentation might ultimately diminish quality of care.

Reviewer #3: 

I found this paper interesting to read but would have liked to have seen more detail on the methodology. I also have concerns that the expert panel was not representative in that it consists of experts more likely to hold positive views about open notes. I wonder if the findings would have been very different if a more representative expert panel had been used.

Response: We thank the reviewer for his or her helpful comments. We have now included more detail on the Delphi methodology on p4 including new references:

“First developed by the Rand Corporation in the 1950s, Delphi polls are now an established qualitative methodology for exploring the consensus views of experts in an emerging field.[15,16] Delphi polls rely on non-probability sampling and studies show they offer more accurate predictions than other forecasting techniques.[17–19] The approach is designed to anonymously pool the views and predictions of a purposive sample of identified experts who are invited to answer a series of open-ended questions on a focused topic. Participants are next required to reassess their initial judgements, in light of aggregate group feedback, until consensus is obtained. While some Delphi polls employ the use of face-to-face meetings to help establish areas of agreement, more robust approaches employ anonymous, iterative techniques to avoid the influences of group think or dominant personalities.[15,18] Delphi polls therefore have clear advantages over focus groups owing to participants’ anonymity and have been used extensively in healthcare policy and management, including in psychiatry.[18,20,21]”

We note that, because Delphi polls using purposive sampling techniques, representativeness is neither intended nor assured. In addition, our experts all had knowledge and experience of the practice of open notes in mental healthcare. However, we do agree that there is a question about representativeness and have amended the text to reflect this limitation. On page 5 we added the clarification:

“We defined expertise as individuals with experience as: […] and/or, academic informaticians, who had published significant contributions within the field of health informatics and patient access to clinical records including authors of publications skeptical about the innovation.”

On page 19 we added:

“In addition, we cannot exclude the possibility that, even though clinicians and chief medical information officers had knowledge and experience of opening mental health notes, some individuals might already have been more positively disposed to the practice, influencing results.”

Minor points

P1, lines 7, 9 and 11 – I would put a comma after ‘round 1’, ‘round 2’, and ‘round 3’.

Response: This has been deleted (see response to Reviewer 2).

P3, para 2, line 3 – I would put a comma after ‘In surveys’.

Response: Now amended.

P4, line 4 – I wonder if the term ‘redact’ might come across as less judgemental than ‘hide’?

Response: Now amended. 

P5, para 2, line 3 – replace ‘clinicians of sharing mental health notes’ with ‘clinicians sharing mental health notes’

Response: Now amended.

P6, para 2, line 2. Would languages other than English have been made available if required?

Response: No, the survey was only made available in English as it was predicted that all prospective experts on sharing mental health notes (who are based mostly in the USA and Sweden, as well as some in England) would be fluent in the language.

P9, Table 1. Could “Age range, y” be replaced with “Age range, years” ?

Response: Now amended.

P15, 4th line from bottom – as before would ‘redacting’ come across as less judgemental than ‘hiding’?

Response: We retained this use of “hiding” since the participants used this phrasing in the open responses. We concede that this wording might prove challenging depending on the perspective of, for example, clinicians versus patients, since “redacting” might also sound jargonistic from a patient perspective. To balance this, we have included “redact” in the Introduction of the paper and retained “hiding” to reflect the wording of the participants, in the Results and Discussion.

P16, line 2 – replace “medication, adherence” with “medication adherence”

Response: Now amended.

Specific concerns

P5, ‘The expert panel’ – The authors describe how individuals with expertise included clinicians with experience of sharing mental health notes with patients and CMIOs of organizations that offer open notes to patients. This would seem to have introduced a bias into the sample in that clinicians and CMIOs who may have actively decided not to offer open notes were not represented?

Response: This is a good point: please see response to comment above, and amendments in the text.

P11, last sentence. The authors mention that thematic analysis was employed, on p7 the authors reference Braun and Clarke. I would like to know a little more information about the epistemological perspective taken by the authors and to see some evidence of reflexivity. I feel there could be more detail on the thematic analysis – the authors mention that 2 individuals were responsible for identifying themes and subthemes – how was this done and what procedures were in place when these individuals disagreed?

Response: Owing to the limitations of the data-set (brief comments, or sentence fragments) full thematic analysis was not applicable but we have now added further elaboration and a reference to that effect. We have also offered details of the background of the authors involved in coding. Finally, the sentence referring to disagreement was inaccurate and has been elided. We have now added on page 7:

“Owing to the limitations of the data-set (brief comments or sentence fragments), full thematic coding was not applicable.[25] Coding was conducted by CB and independently reviewed by JT and MH and subsequent refinements were made. These individuals were selected for their diverse epistemological backgrounds: CB is a philosopher of medicine and healthcare ethicist from the UK, JT is a psychiatrist with experience of sharing mental health notes with patients in the US, and MH is a healthcare informaticist and implementation scientist based in Sweden.”

Pages 12-14. Whilst I can see the value of a thematic analysis of the questions generated by the expert panel, the data would seem to lend itself quite well to a factor analysis. Did the authors consider a factor analysis and if so, what were their reasons for not conducting one?

Response: We did not include factor analysis since this does not form part of Delphi methodology which is a form of qualitative research and we have now offered an enhanced description of Delphi in response to the reviewer’s previous comment.

P16 – It would be good to see some consideration of the difficulties inherent in having two versions of a record when elements of the record have been redacted for the patient view. How can this be managed in practical terms so that clinicians can still see the full record?

Response: While we note that the panelists did not raise the issue of shadow records or redaction of aspects of the record, the reviewer makes an excellent point. We have now added this issue to the discussion. On page 19 we added: 

“Finally, it remained unresolved when redaction of aspects of mental records from patients might be appropriate, and future research should elaborate on how best this can be practically and ethically managed for both patients and clinicians.”

6. PLOS authors have the option to publish the peer review history of their article (what does this mean?). If published, this will include your full peer review and any attached files.

Do you want your identity to be public for this peer review? For information about this choice, including consent withdrawal, please see our Privacy Policy.

Reviewer #1: Yes: Richard Fitton

Reviewer #2: Yes: Ray Jones

Reviewer #3: No

---

## [Editor Report · Decision Letter 1]

17 Sep 2021

The Benefits and Harms of Open Notes in Mental Health: A Delphi Survey of International Experts

PONE-D-21-12428R1

Dear Dr. Blease,

We’re pleased to inform you that your manuscript has been judged scientifically suitable for publication and will be formally accepted for publication once it meets all outstanding technical requirements.

Kind regards,

Umberto Volpe

Academic Editor

PLOS ONE
---

## [Editor Report · Acceptance letter]

30 Sep 2021

PONE-D-21-12428R1 

The Benefits and Harms of Open Notes in Mental Health: A Delphi Survey of International Experts 

Dear Dr. Blease:

I'm pleased to inform you that your manuscript has been deemed suitable for publication in PLOS ONE. Congratulations! Your manuscript is now with our production department. 

Kind regards, 

on behalf of

Dr. Umberto Volpe 

Academic Editor

PLOS ONE